



# Impacts from Hurricane Sandy on New York City in alternative climate-driven event storylines

Henrique M.D. Goulart[1,2], Irene Benito Lazaro[2], Linda van Garderen[4], Karin van der Wiel[3], Dewi Le Bars[3], Elco Koks[2], and Bart van den Hurk[1,2]

[1]Deltares, Delft, The Netherlands
[2]Institute for Environmental Studies, VU University Amsterdam, The Netherlands
[3]Royal Netherlands Meteorological Institute (KNMI), De Bilt, The Netherlands
[4]Institute of Coastal Research - Analysis and Modelling, Helholtz-Zentrum Hereon, Geesthacht, Germany

**Correspondence:** Henrique M.D. Goulart (henrique.goulart@deltares.nl)

**Abstract.** High impact events like Hurricane Sandy (2012) significantly affect society and decision-making around weather/climate adaptation. Our understanding of the potential effects of such events is limited to their rare historical occurrences. Climate change might alter these events to an extent that current adaptation responses become insufficient. Furthermore, internal climate variability in the current climate might also lead to slightly different events with possible larger societal impacts.

5  Therefore, exploring high impact events under different conditions becomes important for (future) impact assessment. In this study, we create storylines of Sandy to assess compound coastal flooding on critical infrastructure in New York City under different scenarios, including climate change effects (on the storm, and through sea level rise) and internal variability (variations in the storms intensity and location). We find that 1m of sea level rise increases average flood volumes by 4.2 times, while maximised precipitation scenarios (internal variability) lead to a 2.5-fold increase of flood volumes. The maximised precipita-

10  tion scenarios impact inland critical infrastructure assets with low water levels, while sea level rise impacts fewer coastal assets though with high water levels. The diversity in hazards and impacts demonstrates the importance of building a set of relevant scenarios, including those representing the effects of climate change and internal variability. The integration of a modelling framework connecting meteorological conditions to local hazards and impacts provides relevant and accessible information that can directly be integrated into high impact events assessments.

## 1 Introduction

Coastal cities face significant exposure to storm-induced compound coastal flooding (Wahl et al., 2015; IPCC, 2022; Woodruff et al., 2013; Dullaart et al., 2021). In the context of storms, compound coastal flooding often involves heavy precipitation and high storm surges (Wahl et al., 2015; Bevacqua et al., 2020). Impacts of coastal storms include fatalities and damage to buildings and critical infrastructure (CI) (Hallegatte et al., 2013; Chang, 2016; Hall et al., 2019). A recent example of a high-

20  impact event is Hurricane Sandy, which struck the East Coast of the United States of America (US) in October 2012. Coastal floodings in New York city (NYC) disrupted several CI systems, impacting millions of people (SIRR, 2013; Kunz et al., 2013).



In the aftermath of the event, and similarly in extreme events around the world, society was activated to reduce vulnerability to similar events in the future (Rosenzweig and Solecki, 2014).

Climate change is projected to increase Tropical Cyclone (TC) precipitation rates and average intensity (Knutson et al., 2020), while rising sea levels are expected to exacerbate the impacts of coastal flooding (Nicholls and Cazenave, 2010; Hallegatte et al., 2013; Hinkel et al., 2014; Oppenheimer et al., 2019). However, significant uncertainties remain regarding the influence of climate change on individual events at regional scales, as global statistics and thermodynamic arguments may not fully apply to local climatological situation (Stott et al., 2016; Gutmann et al., 2018; Trenberth et al., 2018; Shepherd, 2019). Future projections of sea level rise (SLR) also carry significant uncertainties, especially regarding the timing of core processes, such as dynamical ice loss from Greenland and Antarctica (IPCC, 2021; Le Bars, 2018).

Besides climate change, internal variability within the climate system has a defining role in causing specific high-impact events (Deser et al., 2012; Schwarzwald and Lenssen, 2022; Goulart et al., 2023; Hamed et al., 2023). For storms specifically, previous studies have noted that frequency, tracks, and landfall positions exhibit large internal variability (Done et al., 2014; Mei et al., 2015; Bony et al., 2015). Considering impacts rather than meteorological conditions, internal variability has been identified as a major driver of differences, surpassing differences between emission scenarios (Done et al., 2018).

High-impact events, like Sandy, carry great significance for society and societal decision-making, as evident from NYC's intensified climate adaptation plans in response to Sandy (SIRR, 2013; Aerts et al., 2013; Rosenzweig and Solecki, 2014) and in policy-making processes elsewhere around the globe (Gerritsen, 2005; Martinez et al., 2019; Bartholomeus et al., 2023). However, our understanding of the potential impacts of these events is limited to their rare historical occurrences. The impacts of similar events in the future will be different, due to the combination of climate change and internal variability (Otto et al., 2018; Goulart et al., 2021, 2023), potentially hindering the effectiveness of adaptation measures based on historical events and outcomes (Mechler et al., 2010; Haasnoot et al., 2020; Bartholomeus et al., 2023). To comprehensively assess the risk of high impact events, it is essential to consider both climate change and internal variability, and develop a range of relevant and distinct scenarios (Sutton, 2019; Lehner and Deser, 2023). Storylines, plausible self-consistent developments of climatic events (Shepherd et al., 2018), have been previously used for impact assessment applications (van den Hurk et al., 2023a). For example, Qiu et al. (2022) combined a historical TC with global warming projections, sea level rise, and riverine flood to stress-test compound floods in China's Pearl River Delta, and Koks et al. (2023) combined historical high impact storms with different scenarios of sea level rise and future socioeconomic developments to assess coastal flooding damages on critical infrastructure and explore possible adaptation solutions.

In this paper, we employ an event based storyline approach to explore alternative realisations of Sandy and to assess the societal impacts of these alternative events. We study the unfolding of hurricane Sandy under different conditions using a model chain including climate, compound flooding and impact components to establish a clear connection between cause and effect and to obtain a comprehensive understanding of the potential implications associated with these events (Shepherd et al., 2018; Sillmann et al., 2020). We build storylines of Sandy, combining spectrally nudged storyline data, sea level rise scenarios and storm track manipulation, which together account for the effects of climate change and internal variability (Figure 1a). Our modelling framework combines multiple models that cover the chain of events from the meteorological and climatological





characteristics to their flood hazards and to the resulting societal impacts (Figure 1b). As impact metric, we focus on the exposure of buildings and critical infrastructure (CI) assets on the coast of the NYC metropolitan region.

### a) Alternative event storylines

Climate scenarios      Sea level rise (SLR) scenarios      Maximised precipitation (MP) scenario

*Section 2.2.1*      *Section 2.2.2*      *Section 2.2.3*

### b) Modelling framework

**Figure 1.** a) Illustration of the scenarios considered in this study: On the left, the spectrally nudged storylines with different levels of global warming; middle is the sea level rise (SLR) scenarios, and right the maximised precipitation (MP) scenario. b) Modelling framework connecting meteorological and climatic conditions, such as wind speed and mean sea level pressure (MSLP), to flood modelling and critical infrastructure (CI) exposure

## 2 Data and methods

### 2.1 Case study

Sandy began as a tropical depression in the southwestern Caribbean Sea on October 22, 2012, and intensified while moving northward. It peaked as a Category 3 hurricane over Jamaica and Cuba with wind speeds of 50 m/s and a minimum pressure of 954 hPa. The storm caused heavy rainfall and flooding in multiple countries. It encountered a blocking high and a low-pressure



system, which halted its northward movement, and caused it to turn westward and intensify again (Hall and Sobel, 2013; Kunz
et al., 2013; SIRR, 2013).

On October 29, 2012, Hurricane Sandy hit the US East Coast, causing unprecedented coastal flooding in the NYC metropolitan region. Most of the flooding was caused by a high storm surge, but inland precipitation also modestly contributed to the flooding (SIRR, 2013; Kunz et al., 2013; Yates et al., 2014). The event resulted in 43 fatalities and caused US$19 billion in damage in NYC alone (Kunz et al., 2013). Power outages affected 21.3 million people due to cascading effects in the power
system (SIRR, 2013; Kunz et al., 2013; Yates et al., 2014). In this work, we focus the analysis of the impacts of Sandy on the coastal area surrounding the NYC metropolitan region, including parts of New Jersey and Connecticut, as seen in Figure SI A1.

## 2.2 Alternative event storylines

### 2.2.1 Climate scenarios: spectrally nudged storylines

Spectral nudging (von Storch et al., 2000) can be employed to recreate global historical climate events (Schubert-Frisius et al., 2017). We use the event based spectrally nudged storylines dataset from van Garderen et al. (2021), created using the general circulation model (GCM) ECHAM version 6.1.00 (Stevens et al., 2013). In this setup, the large-scale free atmosphere (minimum wavelength of 2000km at altitudes above 750 hPa) is spectrally nudged towards the atmospheric divergence and vorticity derived from the NCEP R1 reanalysis data (Kalnay et al., 1996) to reproduce historical climate events. By nudging
the modelled atmospheric large-scale patterns towards reanalysis, the model simulations are reproductions of historical large-scale weather events, though leaving small-scale processes and the dynamics of the lower atmosphere to respond freely (van Garderen and Mindlin, 2022). This latter point enables the model to respond to different climatological background states which have been prescribed to the model. ECHAM6 is the atmospheric component of the MPI-ESM coupled model (Tebaldi et al., 2021) used in the sixth coupled model intercomparison project (CMIP6). It has a T255 horizontal spectral resolution
and 95 vertical levels (T255L95). More details on the ECHAM6 model and the spectral nudging technique are available on Schubert-Frisius et al. (2017).

The spectrally nudged storylines dataset consists of three climatic worlds governed by different global warming levels (van Garderen and Mindlin, 2022):

- a counterfactual scenario assuming pre-industrial global temperature values (13.6°C, referred to as "PI")

- a present day scenario, in which global temperature is prescribed from observed conditions in 2010 (14.28°C, referred to as "PD"), and

- a counterfactual scenario assuming a global temperature 2°C above the pre-industrial value (15.15°C, referred to as "2C").

The different climatic scenarios were created by modifying model SSTs and greenhouse gas (GHG) concentrations. For the
PD simulations, SSTs and sea ice concentrations were obtained from the NCEP R1 reanalysis data, and the GHG forcing ($CO_2$,





CH4, N20 and CFC's) was based on observed values (Meinshausen et al., 2011). For the PI simulations, the SST climatological warming pattern (calculated as the 2000-2009 average of CMIP6 historical simulations minus the average of CMIP6 PiConntrol simulations) was subtracted from the NCEP R1 reanalysis pattern, GHG forcing was based on observed concentrations from the year 1890. For the 2C simulations, SSTs and GHG values were obtained from outputs of the MPI-ESM model using the

Shared Socioeconomic Pathway (SSP) 5-8.5 scenario (O'Neill et al., 2016) between 2044 and 2053, which correspond to a global temperature of 2°C above pre-industrial levels. All simulations use the same land use and aerosol forcing. Each climatic scenario has three members (generated using different starting spin-up times, see van Garderen (2022)) which are used to investigate the consistence of forced changes and the influence of internal variability on the local/regional conditions within the large-scale nudged events. As such we investigate three alternative realisations of Sandy (internal variability) within three

distinct climate scenarios (climate change: PI, PD, 2C). More details on the spectrally nudged storyline datasets are available on van Garderen et al. (2021) and van Garderen (2022).

### 2.2.2 Sea level rise scenarios

We explore the consequences of the Sandy landfall for different SLR scenarios (Figure 1a), derived from the sixth assessment report (AR6) from the Intergovernmental Panel on Climate Change (IPCC) (IPCC, 2021). They correspond to local estima-

tions of SLR for NYC under a global 2ºC warming, considering only processes for which projections can be made with at least medium confidence (IPCC, 2021). The estimations result from multi-model projections with a global mean temperature increase between 1.75ºC and 2.25ºC in 2080-2100 with respect to pre-industrial levels. They also include different timeframes to account for the temporal variability in SLR for the same temperature increase (IPCC, 2021). The median SLR values in NYC are 71cm for 2100 (referred to as SLR71 scenario) and 101cm for 2150 (referred to as SLR101 scenario). These increases are

relative to local mean sea level from 1995-2014, which account for the effects of local land subsidence. The SLR scenarios are combined with the water levels from Section 2.3.1 to compute flood levels in the alternative realisations of Sandy under SLR.

### 2.2.3 Maximised precipitation scenarios

As discussed previously, the exact tracks of hurricanes are subject to substantial internal variability of the climate system (Done et al., 2014; Mei et al., 2015; Yamada et al., 2019), with varying consequences for society. It can be considered physically

plausible that Sandy could have made landfall in a slightly different, but nearby, location, leading to potentially (very) different impacts. Considering this uncertainty, we explore worst-case realisations resulting from internal variability to obtain impact-relevant outcomes (Sutton, 2019; Schwarzwald and Lenssen, 2022).

We manipulate each simulated storm track from Section 2.2.1 to create another alternative realisation of the event (Figure 1a). These track manipulations are designed to explore the consequences of maximised precipitation over NYC, due to

alternative landfall locations. For that, we calculate the precipitation sum aggregated in a zone of 4 degrees surrounding the storm center from 9 hours before landfall to 9 hours after landfall in the US. We identify the grid cell on land with the highest cumulative precipitation during this time window, and subsequently realign the entire storm track horizontally (shifting storm





track latitudes and longitudes) to ensure that this precipitation maximum is located above the study area (SI Figure A2). We refer to these projections as Maximised Precipitation (MP) scenarios.

We use a tracking algorithm based on Baatsen et al. (2015) and Bloemendaal et al. (2019). For every time step, the initial eye position of the storm is determined by the historical storm data from the IBTrACS dataset (Knapp et al., 2010). We then find the location of the maximum vorticity within a 5°x5° box around the eye position and update the location if it has a lower mean sea level pressure (MSLP) than the initial eye position. After that, we determine the minimum MSLP position within a 2.5°x2.5°box around the updated eye location and select it as the final eye position.

## 2.3    Modelling framework

### 2.3.1    Tides and storm surges modelling

Tides and storm surges are dynamically simulated using the Global Tide and Surge Model (GTSM) v4.1 (Muis et al., 2020). GTSM is a calibrated global hydrodynamic model based on Delft3D Flexible Mesh (Kernkamp et al., 2011). It has an unstructured grid with varying resolution, finer near the coasts (2km in this study) and coarser in the deep ocean (25km). We use data

from the General Bathymetric Chart of Oceans (GEBCO) 2014 dataset (GEBCO, 2014) with a 30 arc seconds resolution as input for the bathymetry. GTSM uses tide-generating forces and simulates storm surges based on wind speed and atmospheric pressure forcings. The combined effects of tides and storm surges determine the total water levels. We force the model with mean sea level pressure (MSLP) and wind speed data from Sections 2.2.1 and 2.2.3 to generate water levels in the study area.

### 2.3.2    Compound coastal flooding modelling

We simulate the compound flooding on the study area with the Super-Fast INundation of CoastS (SFINCS) Model (Leijnse et al., 2021). SFINCS is a reduced-physics solver used to simulate compound (pluvial, fluvial and coastal) flooding in coastal areas. It solves simplified equations of mass and momentum to simulate overland flow in two dimensions, accurately estimating flooding while being computationally efficient (Leijnse et al., 2021). SFINCS incorporates physical processes such as spatially varying friction and infiltration, and has been used previously to assess the compound flood impact of TCs (Leijnse et al., 2021;

Sebastian et al., 2021; Eilander et al., 2023b). A full description of the model is available at Leijnse et al. (2021).

In this study, we setup the SFINCS model and process the input data using the Python package HydroMT (Eilander et al., 2023a). For surface elevation we use modern and high resolution datasets publicly available: The Continuously Updated DEM (CUDEM, both 1/9 and 1/3 arc-second resolution) (Amante et al., 2023) and the 0.3048m resolution NYC Topobathy Lidar DEM (OCM Partners, 2017) for the coastal topography and bathymetry of the NYC region. For the areas where those datasets

are not available, we use the global datasets FABDEM (Hawker et al., 2022) and GEBCO (GEBCO, 2014) to fill missing data. The roughness coefficients used in our models are obtained from the Copernicus Global Land Service (Buchhorn et al., 2020) and infiltration data from the GCN250 dataset (Jaafar et al., 2019). We run SFINCS at 50m resolution and we force the model with precipitation and water levels from the previous steps to obtain flood maps of the study area. We also simulate the events with precipitation and coastal water levels separately to estimate their contribution into compound flooding.



### 2.3.3 Societal impact modelling: critical infrastructure data and exposure

To assess the potential societal consequences of alternative realisations of hurricane Sandy, we perform an exposure analysis of the built environment, accounting for both CI assets and buildings. The assessment involves overlaying geospatial information of buildings and assets with flooding maps, which allows evaluating the built environment exposure to flooding.

Apart from all buildings within the study area, our study includes seven major CI systems: energy, transportation, telecommunication, water, waste, education, and health, as identified by Nirandjan et al. (2022). To obtain the necessary CI data, we rely on the widely accessible OpenStreetMap (OSM) database (Haklay and Weber, 2008). This source has been utilised in multiple studies (Koks et al., 2019; Nirandjan et al., 2022; Koks et al., 2023; Liu et al., 2023), demonstrating its suitability for our purposes. Information gaps exist within OSM and the level of completeness varies substantially across the world (Herfort et al., 2023). For NYC, the fraction of buildings included has been estimated to between 55 (Zhou et al., 2022) and 80% (Herfort et al., 2023). For CI assets, only very limited coverage estimates are available. Above 90% of all roads in NYC is estimated to be included (Kazakov et al., 2023).

Different CI assets may exhibit varying responses to distinct flood levels. Unfortunately, comprehensive information regarding the vulnerability of different assets to specific flood levels is limited (Zio, 2016). Consequently, we analyse exposure divided in three water level categories: low (0.05m-0.5m), medium (0.5m-1m), and high (>1m) water levels. This approach allows an assessment of the potential flood impact on the built environment.

## 3 Results

### 3.1 Alternative meteorological event realisations and climate change scenarios

We evaluate the potential impact of varying global warming levels and internal variability on Sandy. All nine runs (three storylines, three members each) have alternative realisations of hurricane Sandy that are close to the observed event. The storm tracks begin over the Caribbean Sea, move along the US east coast, and turn towards the US coast (Figure 2a). All tracks have landfalls slightly north of the observed landfall, but their mutual differences are minor, and no coherent response of track position to the imposed warming levels is detected, which is to be expected when using spectrally nudged data (von Storch et al., 2000). Meteorological features match the observed event well in the region of interest (Figure 2b-c). The simulations miss the peak intensity of the storm over the Caribbean (between 24 and 26 Oct). This discrepancy is also present in other datasets, such as the spectrally nudged global historical dataset (ECHAM_SN) (Schubert-Frisius et al., 2017) and two modern reanalysis datasets, ERA5 (Hersbach et al., 2020) and MERRA-2 (Gelaro et al., 2017) (Figure SI A3).

When averaged across each climate scenario, MSLP and maximum wind speeds show no significant changes (Figure 2b-c). Peak precipitation rates during October 24th to 26th show significant gains due to climate change (Figure 2d where PI members (blue) lie below the 2C members (brown) for day 24-26), with an ensemble mean 14% increase from PI to PD and a 5% increase from PD to 2C. Such a systematic response is absent in the precipitation rates produced during landfall over the study area. Increases in precipitation generally occur for the most extreme precipitation rates (Gutmann et al., 2018), but by this point, the





**Figure 2.** Tracks of alternative realisations of Sandy (a). Timeseries of the alternative realisations for minimum MSLP (b), maximum wind speed (c) and mean precipitation around the storm eye (d). PI, PD and 2C are represented by blue, orange, and red, respectively, while the 3 members are represented by solid, dashed and dotted lines. The grey dashed-dotted line represents the observed values for track, MSLP and wind speed.



storm is transitioning into an extratropical (ET) storm, resulting in overall lower precipitation intensity. Thus, over the study area, we find no significant climate change signal in Sandy's alternative realisations. There are however considerable differences across all simulations, related to internal variability, making them highly relevant for further exploration. We therefore focus our subsequent analyses on exploring this variability and its impacts on the study area, without attributing changes to climate change.

## 3.2 Flood hazards

The variability among the alternative realisations of Sandy results in a wide range of flooding events (see Figure 3a). Some realisations exhibit up to 3.5 times more flooding volume in the study area than others. We illustrate the compound nature of the flood events produced by the storms, as both precipitation and surge contribute to the flooding volume in all cases. The varying characteristics of rainfall patterns and storm surge levels lead to a diversity in flood scenarios, some of which are primarily driven by precipitation, while others are dominated by storm surge. The compound effects of the storm indicate that for the flood volume storm surge and inland precipitation have minimal interaction, closely resembling the simple combination of each component modelled separately. In some coastal areas and wetlands the compound effects lead to a relatively small reduction in flood volume compared to the direct aggregation of precipitation- and surge-only events.

The spatial distribution of flood hazards exhibits distinct characteristics for surge-dominated and precipitation-dominated events. Surge-dominated flood events primarily affect coastal areas and result in high flood levels (Figure 3b). Conversely, precipitation-dominated flood events have a broader spatial extent, reaching into inland areas, but typically exhibit lower flood depths (Figure 3c).

## 3.3 Evaluation of sea level rise scenarios

Next we investigate the flood hazards of the alternative realisations of Sandy for the different SLR scenarios. All simulations present an increase in flood volume as sea levels rise (Figure 4a). SLR71 results in an average increase in flood volume by approximately threefold, ranging from 2.2 to 3.7 times higher than the corresponding baseline simulations. SLR101 increases flood volume by 4.2 times, ranging from 3 to 5.6 times higher than the baseline simulations. For each SLR scenario, higher increases occur for higher initial flood volume caused by surges (see bars, dashed lines and slope coefficients in Figure 4a). Most of the flood volume increase occurs on the coast (Figure 4b).

## 3.4 Evaluation of maximised precipitation scenarios

The effects of manipulating the tracks of the alternative realisations of Sandy to maximise the precipitation over the study area results in significant flood volume increases. The manipulated storm tracks are depicted in Figure SI A4. Figure 5a shows that the average flood volume increases by a factor of 2.5, ranging from 1.6 to 3.7 times the volume in the corresponding events with original tracks (baseline). Additionally, there is significantly less variation in flood volume in the manipulated storms, with all realisations but one being in the range $4 +/- 0.3 \times 10^8 \, \mathrm{m}^3$, while the baseline shows a range between 0.6 and 2.5 $10^8 \, \mathrm{m}^3$.







**Figure 3.** (a) Compound flood volumes over the study area for each event (darkred dots), compared to the corresponding univariate surge or rain induced flood hazards represented by bars. Lightblue and pink bars indicate surge- and precipitation-driven floods, respectively. Flood hazard maps for the largest surge-dominated (b) and precipitation-dominated (c) flooding volume storylines.



**Figure 4.** a) Compound flood volumes for baseline (lightblue circle), 71cm SLR (blue X) and 101cm SLR (darkblue square). The corresponding bars show the increase in flood volume for each scenario due to storm surge only (the difference between bar and symbol is the flood volume by precipitation). Dashed lines show the trends and regression coefficients for each SLR scenario. b) Difference in flood between SLR101 and baseline scenarios for the same event.





Therefore, although most realisations produce similar precipitation volumes, the landfall position determines the flood volumes in the study area. The increased flood volumes due to the MP scenario occur extensively across the study area, but coastal areas

show a moderate decrease in inundation levels (Figure 5b). This is due to the MP scenario setup, where inland precipitation over the study area is prioritised over wind speed and MSLP contributions to local surge levels, which decrease consequently.

## 3.5   Critical Infrastructure exposure

In this section, we assess the exposure of CI and buildings to flood hazards caused by the alternative realisations of Sandy. The MP scenario results in the highest number of flooded assets, typically 5 times the baseline (Figure 6a). Following this is

the SLR101, which shows a 2.3-fold increase, and the SLR71, with a 2-fold increase. This is due to the extensive reach of precipitation. As a result, most of the increase in the MP occurs in the low water level category (5.9 times), while SLR71 and SLR101, surge dominated scenarios, increase 1.3 and 1.4 times, respectively. For medium water levels, SLR71 (2.9 times) and SLR101 (3.3) show higher increases than MP (2.8). For high water levels, SLR101 shows the highest increase, 8.7 times the baseline. SLR71 follows with a 4.5-fold increase, and the MP, with a 2.6-fold increase. Among the CI systems, buildings

and roads show the highest number of exposed assets (Figure SI A5). SLR101 leads to the largest increase in number of flooded assets across the CI systems compared to the baseline (Figure 6b). Power infrastructure has the highest increase in exposure, though education, telecom and wastewater systems also see considerable increases. The impacts of SLR scenarios are predominantly driven by storm surges, impacting mostly coastal assets with high water levels (Figure 6c). In contrast, the MP scenario, where flooding is primarily driven by precipitation, has a spatially extensive impacts on assets, but at low water

levels (Figure 6d).

## 4   Discussion

In this paper, we use storylines to develop alternative yet plausible realisations of hurricane Sandy as it made landfall in NYC, accounting for effects of climate change and internal variability. Our objective is to increase our understanding of Sandy and its impacts beyond the single historical occurrence and to obtain climate information that is relevant for risk assessment (Sutton,

2019). We develop a modelling framework spanning meteorological conditions, compound flooding and CI flood exposure, allowing for a comprehensive analysis of the event onset and consequences.

Compound floods on the study area over the range of alternative realisations result from a combination of storm surges and precipitation. However, the contribution of each hazard differs substantially among the realisations and scenarios explored, demonstrating the importance of compound thinking in coastal flooding (van den Hurk et al., 2023b). Surge dominated floods

concentrate in coastal regions with high water depths, while precipitation dominated floods cover widespread land areas with shallow water depths. We calculate the exposure of critical infrastructure assets to floods, as a proxy of potential societal impacts of alternative realisations of Sandy. Exposed CI assets vary according to the prevailing flood hazards of each event: precipitation dominated events result in a considerable increase in the number of exposed assets, mostly at low water levels. Conversely,





**Figure 5.** a) Similar to Figure 4, but for the maximised precipitation (MP) scenario. Compound flood volumes for baseline (violet circles) and MP (purple crosses). Corresponding bars indicate flood volume due to precipitation only. b) Difference in flood between MP and baseline scenarios for the same storyline.



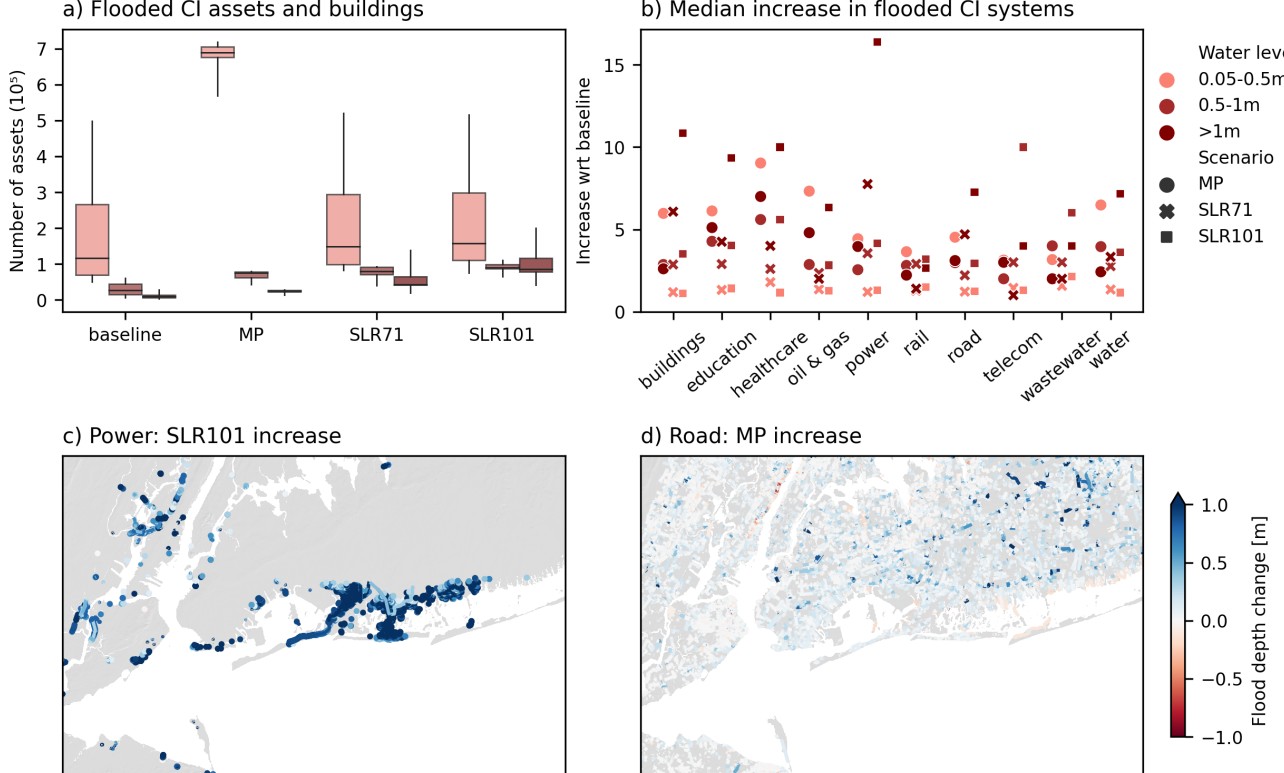

**Figure 6.** a) The total number of flooded CI assets under different water level categories for the baseline, MP, SLR71 and SLR101 scenarios. Low (0.05m-0.5m), medium (0.5m-1m), and high (>1m) water levels are represented by light pink, light red and dark red, respectively. b) The increase in number of flooded CI assets with respect to the baseline. MP, SLR71 and SLR101 are represented by circles, crosses and squares. The difference in exposed assets under various scenarios: c) power assets between the SLR101 and baseline scenarios, d) road assets between the MP and baseline scenarios.

surge dominated events concentrate exposure on coastal assets, particularly at high water levels. The variability in exposed CI

assets illustrates the range of impacts that NYC could have experienced across different (plausible) event unfoldings.

We show SLR substantially increases flood volumes for all alternative realisations of Sandy, with approximately 1m of sea level rise leading to a 4.2-fold increase in flood volume. This is consistent with literature expressing high confidence in the role of sea level rise as a prominent climate change factor explaining tropical cyclone impacts (Lin et al., 2016; Knutson et al., 2020). We manipulate the tracks of the alternative realisations of Sandy to maximise precipitation during landfall over the study

area. This way, we explore the internal variability of the landfall location to search for worst-case scenarios that are relevant for society (Sutton, 2019; Schwarzwald and Lenssen, 2022; Lehner and Deser, 2023). This approach leads to an average 2.5-fold increase in flood volume. In contrast to changes attributed to climate change, which occur over longer timescales, internal



variability applies to present-day climate conditions. This requires a different risk impact perspective and demonstrates the importance of internal variability in quantifying risks of high impact events.

While Sandy becomes wetter during its peak in the Caribbean in the warmer scenarios, we find no significant changes in precipitation over the NYC metropolitan region. Previous studies have found a global increase in precipitation for TCs with climate change (Hill and Lackmann, 2011; Patricola and Wehner, 2018; Knutson et al., 2020), and specifically for Sandy (Yates et al., 2014; Gutmann et al., 2018). The largest increases in precipitation occur generally during extreme precipitation rates (Gutmann et al., 2018), which are not prevalent at Sandy's landfall over NYC. Yates et al. (2014) found higher precipitation
during landfall under 4°C warming but modest increases under 2°C warming. We do not find significant changes for wind speed and MSLP, which could be due to the spectral nudging method where the divergence and vorticity are set to match reanalysis (von Storch et al., 2000; Weisse and Feser, 2003). Yates et al. (2014) also found no significant changes in wind speed and MSLP, while other studies suggest the core pressure of Sandy could decrease in warmer scenarios (Lackmann, 2015; Gutmann et al., 2018). Conversely, we observe significant internal variability between the alternative realizations of Sandy during landfall
over the study area. Given our study aim of exploring societal impacts in alternative realisations of Sandy, we decided to focus on the internal variability of the simulations during landfall rather than at the direct effects of climate change on the entire storm lifetime.

     We rely on the use of one single model for each step in our framework. More robust results can be achieved with a wider selection of models to account for model uncertainty. The spectrally nudged storylines have 3 ensemble members, and while
they are suitable for the purposes of our study, more ensemble members improve results robustness. ECHAM6.1 has an approximate resolution of 0.5 degrees, and studies have shown that higher horizontal resolutions lead to better modelled TCs (Knutson et al., 2020), higher surge heights (Bloemendaal et al., 2019) and better reproduction of precipitation extremes (Prein et al., 2016). However, spectrally nudged model simulations do resolve TC's better than free running simulations (Feser and Barcikowska, 2012; Schubert-Frisius et al., 2017). Our warmest storyline is based on SST and GHG projections of a 2°C
above pre-industrial levels scenario, but due to indirect aerosol influence, the actual temperature increase is 1.55°C, making it a conservative estimation of the climate-change signal (van Garderen and Mindlin, 2022).

     Assessing the risks of high impact events in a changing world requires methods that extend beyond historical observations (Otto et al., 2018; Sutton, 2019; van den Hurk et al., 2023b). Qiu et al. (2022) and Koks et al. (2023) showed that presenting impact changes in coastal flooding through alternative scenarios of historical storms provides clear and accessible information
for decision makers. According to our findings, healthcare decision makers may focus on future asset exposure to surge levels of a future Sandy combined with sea level rise, while road infrastructure decision makers may prioritize the immediate exposure of roads to an alternative Sandy with high precipitation over the study area. Adaptation solutions would differ for each case, for example, sea barriers against surges and expanding urban green spaces for high precipitation. The use of societal-relevant scenarios, including climate change and internal variability scenarios, along with an impact-assessment framework, provides
relevant and accessible information that can be integrated into decision making to achieve effective adaptation solutions (Aerts et al., 2014; Jongman, 2018).



## 5 Conclusion

High impact events affect society and influence decision making. In this study, we create alternative realisations of Sandy to understand the impacts of the event on critical infrastructure over the NYC metropolitan region under different scenarios. The
scenarios are developed to account for the effects of climate change (on the storm, and through sea level rise) and internal variability (random variations in the storms location, intensity and shape in present-day climate). Our framework allows us to simulate all the contributing factors of the event and to disentangle its main components, from driving meteorological and climatological conditions to compound flooding and impacts.

We find that sea level rise is the most consistent climate change component to increase Sandy's flood volumes, with an
average 4.2-fold increase for 1m of sea level rise. However, internal variability, represented by both results from a climate model ensemble and the manipulation of storm's landfall position, also considerably increases flood volumes. For the maximised precipitation scenario, flood volumes exhibit an average increase of 2.5 times compared to the baseline. While all alternative realisations incur in compound floods, the contribution of each flood hazard greatly influences the extent and depth of these events, resulting in distinct impacts on critical infrastructure. Precipitation dominated realisations lead to the highest number
of exposed assets, but mainly at low water levels. Surge dominated events affect mostly coastal assets, with high water levels.

The considerable differences in hazards and impacts demonstrate the potential of building societal-relevant scenarios that provide plausible realisations of a historical high impact event under diverse circumstances. By understanding the potential changes in the event's impacts and their underlining scenarios, decision makers are better informed to make effective adaptation solutions, particularly in a changing climate.

*Code availability.* The code for this experiment is available at: https://github.com/dumontgoulart/sandy_impacts_storylines. SFINCS is available at https://sfincs.readthedocs.io and HydroMT is available at https://deltares.github.io/hydromt/.
For information on GTSM: https://publicwiki.deltares.nl/display/GTSM.

*Data availability.* GEBCO is available at https://www.gebco.net/, FABDEM at https://www.fathom.global/product/fabdem/, CUDEM at https://www.ncei.noaa.gov/access/metadata/landing-page/bin/iso?id=gov.noaa.ngdc.mgg.dem:999919. The spectrally nudged storylines are available upon request to Linda van Garderen.



**Appendix A:  Supplementary information**

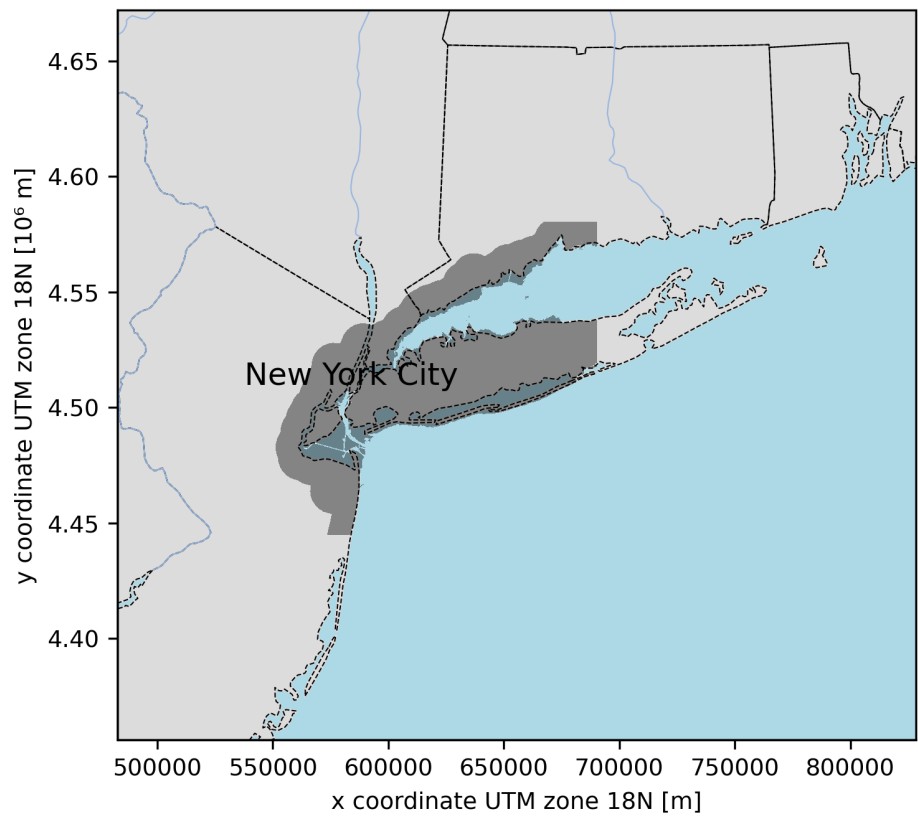

**Figure A1.** Northeast coast of the US and area of analysis (black shaded).



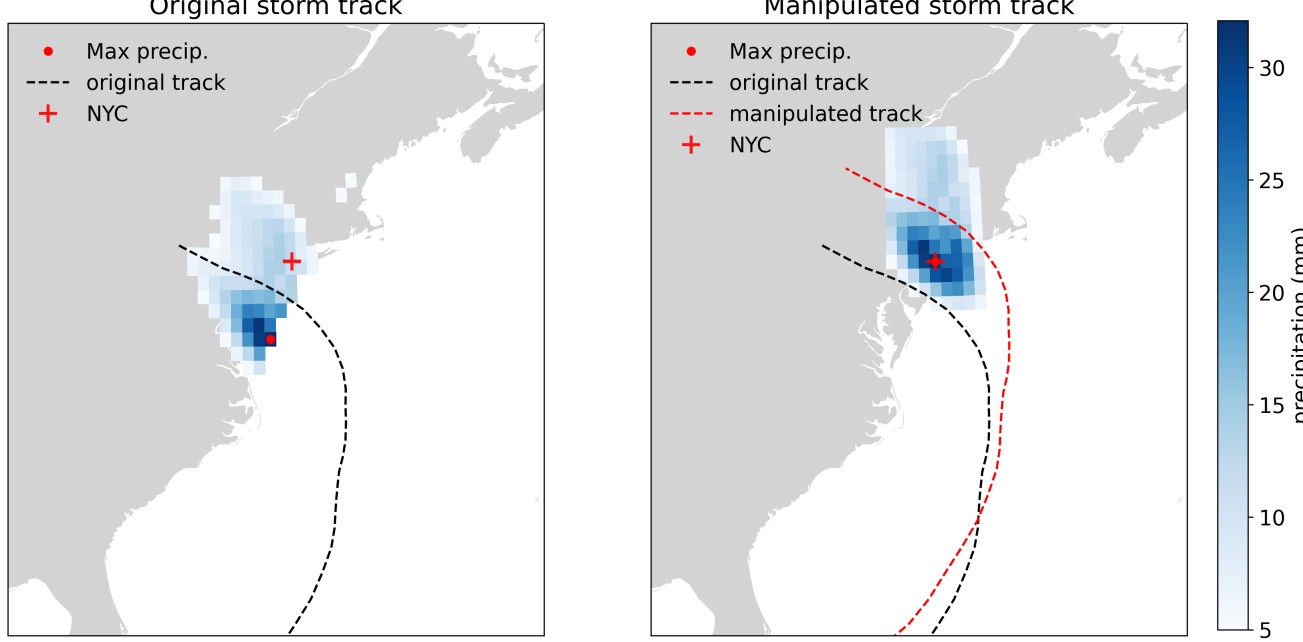

**Figure A2.** Storm track manipulation based on the maximum precipitation location. Blue squares indicate precipitation levels (mm), dashed lines indicate storm tracks and the cross shows the location of NYC.



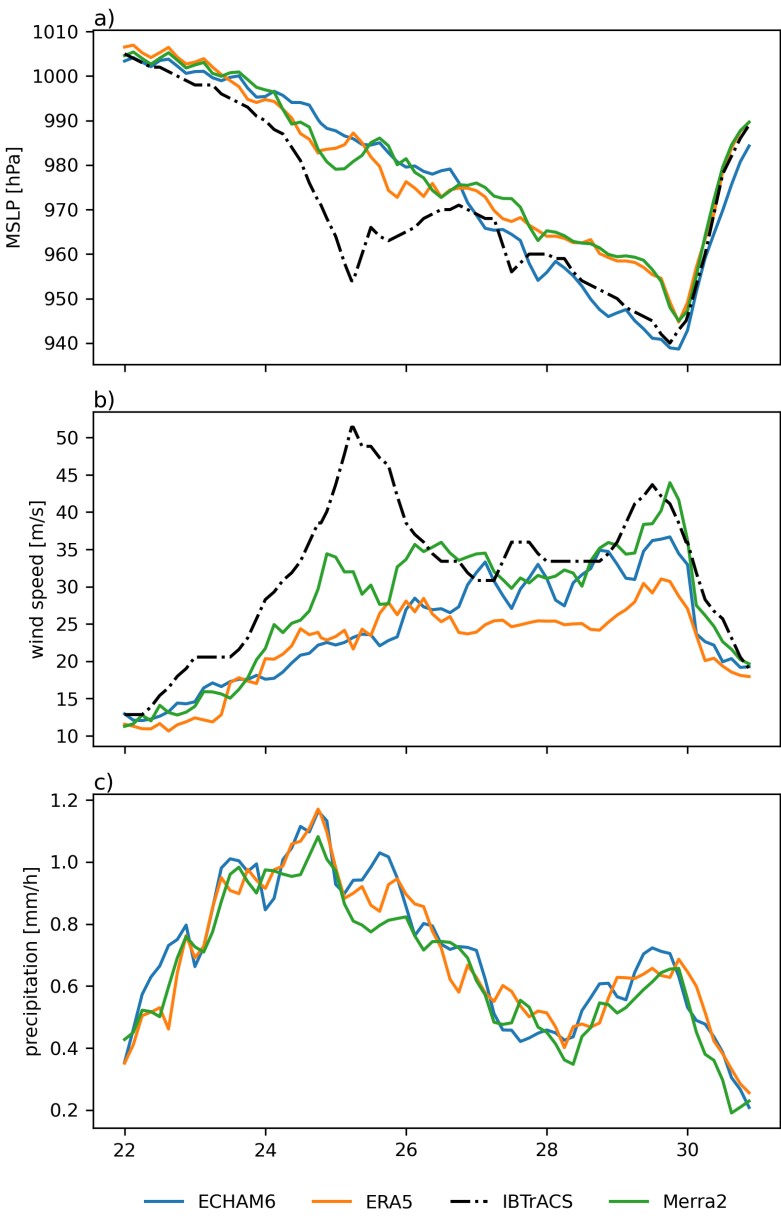

**Figure A3.** Similar to Figure 2. The historical spectrally nudged dataset (ECHAM_SN), ERA5, Merra2 and IBTrACS are represented by blue, orange, green and dashed black, respectively. No IBTrACS information for precipitation.



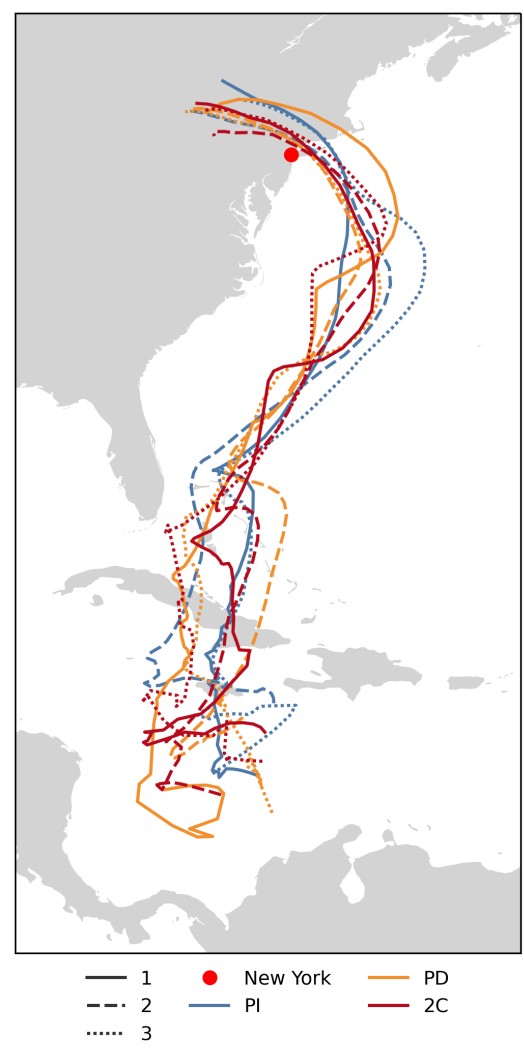

**Figure A4.** Similar to Figure 2a, but for manipulated storm tracks from MP scenario.



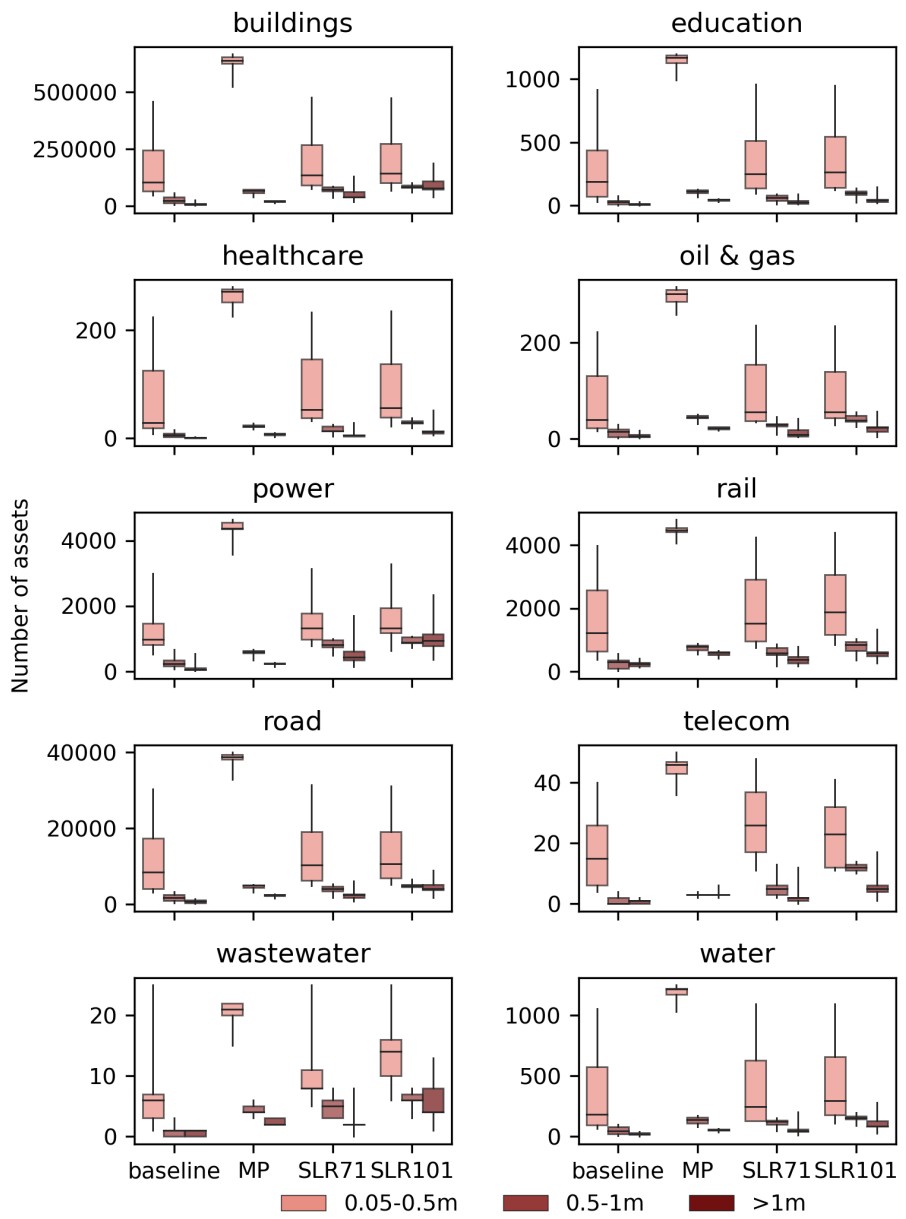

**Figure A5.** Similar to Figure 6, but for different CI systems.



*Author contributions.* HG, KvdV and BvdH contributed to the concept of the study. HG conducted the research and edited the manuscript. LvG, DlB, EK and IBL provided the data. IBL provided the GTSM model and EK the scripts for critical infrastructure. All authors discussed the analysis and results, and revised the manuscript. BvdH, KvdW and EK supervised the work.

*Competing interests.* The authors declare that they have no competing interests.

*Acknowledgements.* This research has been supported by the European Union's Horizon 2020 research and innovation programme under grant agreement No 820712 (project RECEIPT, REmote Climate Effects and their Impact on European sustainability, Policy and Trade). We thank N. Bloemendaal, T. Busker, D. Purnamasari, D. Peregrina Gonzalez, R. Hamed, T. Happe, D. Eilander, T Leijnse, S. Muis and F. Feser for comments on previous versions of the manuscript.



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
