# Peer review of "Compound flood impacts from Hurricane Sandy on New York City in climate-driven storylines"

_EGUsphere, 2023_

## Author Comment (AC1)

**Impacts from Hurricane Sandy on New York City in alternative climate-driven event storylines**

Henrique M. D. Goulart, Irene Benito Lazaro, Linda van Garderen, Karin van der Wiel, Dewi Le Bars, Elco Koks, and Bart van den Hurk

**= Response to the reviewers =**

**Reviewer #1:**

The paper has successfully addressed the issue of estimating the consequences of high-impact-low-frequency events. The research applied a combination of profound models to generate multiple scenarios of global warming, sea level rise, and storm tracks, as well as simulate corresponding flood events. The adoption of multiple stochastic process-based models greatly reduced the inherent uncertainty and arbitrariness of the Storyline Approach. Furthermore, the study managed to include and compare the factors of climate change and internal variability. This further contributes to the understanding of major driving force of high-impact events in the future. With mostly positive feedbacks and pleasant learning experience, there are some minor comments for the current manuscript.

*We would like to genuinely thank the reviewer for their constructive review of our manuscript. In this document we respond to these comments and highlight the modification in the revised text.*

1. Climate scenario constructions. The paper in general well explained how the researchers employed spectral nudging to recreate climate events under various climate conditions. However,

   a. What is the necessity of creating a pre-industrial (PI) climate scenario, instead of building a climate change scenario warmer than 2 degree?

*Thank you for the comment. We agree that exploring warmer scenarios can be beneficial for the analysis of global warming impacts. However, the generation of these scenarios by spectrally nudging GCMs to reanalysis data and then changing the boundary conditions can be quite complex and time consuming. Therefore, the data used in the paper comes from an already existing dataset (van Garderen, 2021) and was not developed for this study. The dataset has the same 3 climate scenarios used in this study (PI, PD and 2C).  The idea for these scenarios is that with the pre-industrial simulation we understand the changes we are already experiencing, and putting those in a climate change context for a near future situation of 2C.*

*We added to the limitations paragraph at the discussion section the following:*

*Line 304: "Our warmest storyline is based on SST and GHG projections of a 2◦C above pre-industrial levels scenario, but due to indirect aerosol influence, the actual temperature increase is 1.55◦C, making it a conservative estimation of the climate-change signal (van Garderen and Mindlin, 2022). Warmer climate scenarios can provide extra insight on the effects of global warming on storms, as seen in Yates (2014) where the strongest precipitation increases occurred for the +4C scenario."*

    b.   More reflection is recommended on the validity of these reconstructed tracks. As the authors pointed out in Figure 2 and line 221, the simulated storm tracks did not well represent the MSLP, highest wind speed, or the variation in flood volume, though they in fact have rather successfully reproduced the situation when the storm hits NYC. Such discrepancies should be better discussed.

*Thank you for the comment, this is indeed an interesting point of discussion. The peak in the TC activity over the Caribbean between 24 and 26 depicted by the observation is missed by the spectrally nudged storylines. But it is also missed by the other two modern and widely used reanalysis datasets (ERA5 and MERRA-2), and by the historical spectrally nudged simulations (ECHAM_SN), as shown in figure A3. When compared to these datasets, the spectrally nudged storylines perform similarly over the study area.*

*According to Hodges et al. (2017), reanalysis products tend to underestimate the peaks in both maximum wind speeds and minimum MSLP (mean sea level pressure). This is likely a consequence of not high enough model resolution and dependence on parameterized processes used in the reanalysis. They also mention that modern reanalysis products show an improvement in reproducing TCs, such as demonstrated with MERRA-2.*

*Ref: Hodges, Kevin, Alison Cobb, and Pier Luigi Vidale. "How well are tropical cyclones represented in reanalysis datasets?" Journal of Climate 30.14 (2017): 5243-5264.*

*Therefore, the discrepancies seen are not particular to the spectrally nudged storylines, but true to all models, and likely due to their limitations in simulating TC peak activities. Conversely, we see that the spectrally nudged storylines perform similarly to the best reanalysis products currently available. When combined to the evidence shown that the results are approximate to the observations over the region of NYC (our study area), we believe the spectrally nudged storylines can be used for the purposes of our study.*

*Based on this discussion, we added to the XX(discussion/results?)XX section:*

*Line 193: "Meteorological features match the observed event well in the region of interest, with some minor underrepresentation of the maximum wind speed (Figure 2b-c)"*

*Line 296: "The simulated storms underrepresent the maximum wind speed and minimum MSLP during the TCs peak over the Caribbean and to a lesser extent during landfall. Similar discrepancies are seen for the other reanalyses tested, indicating the data is within the same range of performance of other reanalyses and models. Peak TC activity is often underrepresented in reanalyses due to limited model resolution and dependence on parametrised processes (Hodges et al., 2017)."*

*In addition, we also added to Figure A3 the range of the spectrally nudged storylines for a better comparison with the reanalyses:*

[Figure]

2. Vulnerability of CI to different water levels. It is well understandable that precise estimation of the fully continuous vulnerability curve of various CI is nowhere to find. Therefore it is a common approach to use a discrete and qualitative impact function. However, it remains quite confusing to me how the authors in Section 3.5 managed to give a quantify the change of impacts. I assume the authors actually assigned a percentage of damage to each level of exposure defined in the paragraph between line 172 and 175. It may be more clear to give this numerical relationship.

*Thank you for the comment. As the reviewer has already pointed out, we adopt a discrete and qualitative approach in separating the water levels because we are analysing multiple CI systems. However, we do not assign percentages of damage to each level of exposure in posterior step. The changes in impacts shown in Section 3.5 and illustrated by Figure 6 a) and b) are the changes in the number of exposed assets for each water level category between the different scenarios in the study. The changes in Figure 6 c) and d) are the differences in water level (m) for the same CI assets between the different scenarios.*

*We believe the text could be better explained, so we rewrote parts of the methods section to improve the clarity of our approach:*

*Line 180: "Different CI assets may exhibit varying responses to distinct flood levels. Unfortunately, comprehensive information regarding the vulnerability of CI assets to specific flood levels is limited (Zio, 2016) and, in particularly, the cost of reconstruction and replacement of CI assets is not available for New York City. Inspired by Koks et al. (2019), we adopt a discrete and qualitative approach by dividing water levels in three categories: low (0.15m-0.5m), medium (0.5m-1m), and high (>1m). This approach allows to quantify the number of exposed assets in each water level category and how it changes under different scenarios, identifying hotspots of impacts, without trying to assign specific monetary value."*

3. Results presenting. Figure 2 on Page 8 could probably have been polished, such that the simulated results of MSLP, wind speed and precipitation in NYC could be highlighted.

*Thank you for the suggestion! We agree that having a better indication of the period that the storm is on the study area Is beneficial for the paper. Therefore, we have updated the figure as follows:*

[Figure]

---

## Author Comment (AC2)

**Impacts from Hurricane Sandy on New York City in alternative climate-driven event storylines**

Henrique M. D. Goulart, Irene Benito Lazaro, Linda van Garderen, Karin van der Wiel, Dewi Le Bars, Elco Koks, and Bart van den Hurk

**= Response to the reviewers =**

**Reviewer #2:**

The authors present an approach for understanding the potential impacts of high-consequence tropical cyclones by generating alternate, realistic tracks of a historical event subjected to natural variability and different climate states. These alternate tracks are then simulated using a high-resolution coastal flood model to investigate how differences in the TC tracks/hazards result in differences in flood depth/extent. The framework is applied to Hurricane Sandy, demonstrating that different potential tracks of Sandy could have resulted in widely different flood dynamics. Overall, the paper is well-written and the storyline approach presented here could be very useful in developing coastal hazard scenarios in support of decision-making. I have several questions/comments that should be addressed to improve the interpretability of the manuscript.

*We would like to thank the reviewer for their positive feedback. We welcome the suggestions proposed. Below we note the revisions done in response to all the suggestions.*

1. Section 2.2: Please add a table summarizing all the scenarios. At some point I lost track of the number of storylines. 3 climate states x 3 internal variability runs x 2 SLR scenarios x 2 precipitation scenarios...?

*Thank you for the suggestion. We agree that a table summarizing all scenarios can improve the clarity of the study. We added a table summarizing all the scenarios as follows:*

*Line 74: "The scenarios used in this work to build alternative event storylines of Sandy are summarised in Table 1 and are explained in the following sections."*

**Table 1.** Table summarising the scenarios considered in this study, the corresponding number of members in each scenario and in which section they are explained.

| Scenario name | Number of members | Name of members | Section |
|---|---|---|---|
| Climate scenarios | 9 (3 climate states x 3 ensemble members per scenario) | Climate states: PI, PD, 2C | 2.2.1 |
| Sea level rise scenarios | 3 | baseline, SLR71, SLR101 | 2.2.2 |
| Maximised precipitation scenarios | 2 | baseline, MP | 2.2.3 |

2. Line 75: need a clear one-sentence description of spectral nudging.

*Thank you for the suggestion. We updated our description of the technique, summarizing the spectral nudging:*

*Line 76: "Spectral nudging (von Storch et al., 2000) is a technique used to recreate historical climate events by forcing the large-scale atmospheric patterns in climate models with reanalysis data while allowing small-scale processes to respond freely (Schubert-Frisius et al., 2017). We use the event based spectrally nudged storylines dataset from van Garderen et al. (2021), created using the general circulation model (GCM) ECHAM version 6.1.00 (Stevens et al., 2013)…"*

3. Section 2.2.1 and 2.2.2: Why are the climatology and SLR scenarios not consistent? They use very different time periods (i.e. 2044-2053 for the storm climatology and 2080-2150 for SLR), which does not make sense to me. I understand that both the climatology projections and SLR projections are taken for a 2 degree C (roughly) scenario. But couldn't the authors take the SLR projections from the same GCM (i.e. MPI model) so that the SLR and climatology are consistent with each other. See Lockwood et al. (2022).

*Lockwood, J. W., Oppenheimer, M., Lin, N., Kopp, R. E., Vecchi, G. A., & Gori, A. (2022). Correlation Between Sea-Level Rise and Aspects of Future Tropical Cyclone Activity in CMIP6 Models. Earth's Future, 10(4).* [https://doi.org/10.1029/2021EF002462](https://doi.org/10.1029/2021EF002462)

*Thank you for the comment. The effects of temperature increase on Sea level rise (SLR) are different than on storms and TCs. SLR processes have large uncertainty regarding the timing of core processes (Dewi Le Bars, 2018), which renders the response of SLR to a certain temperature increase considerably variable. In parallel to this, storylines are not focused on the most probably projection at certain conditions, but rather on exploring plausible possibilities of future scenarios. Therefore, our objective is not to establish the projected SLR at the same time of the 2C climate scenario, but to obtain physically plausible SLR levels for a 2°C warmer world.*

*Because of that, our assumption is that we can compare the climate scenarios and the SLR scenarios at different time periods, as long as they correspond to a physically compatible global warming scenario (following the plausibility principle of storylines). This gives us more possibilities to explore societally-relevant scenarios, such as higher SLR levels for the same thermodynamical conditions. Conversely, restricting our SLR scenarios only to the same time period of the 2C climate scenario of this paper (2044-2053) would restrict the number of scenarios to one, and the extent to which we could assess the potential impacts of SLR.*

*We appreciate the suggestion on using the MPI data from Lockwood et al. (2022). The limitations are that their SLR results shown in the paper (Table 1) are only for Wilmington, NC and only for the time period 2080-2100 at an SSP5-8.5. This makes the comparison physically inconsistent as the corresponding temperature increase is 3.7 °C, and the number of scenarios would be reduced.*

*Le Bars, D. (2018). Uncertainty in sea level rise projections due to the dependence between contributors. Earth's Future, 6, 1275–1291. https://doi.org/10.1029/2018EF000849*

*We updated the methods section to improve our justification on why we use different time periods between the two scenarios and add to our discussion the limitations of considering them independent:*

*Line 111: "We explore the consequences of the Sandy landfall for different SLR scenarios (Figure 1a), derived from the sixth assessment report (AR6) from the Intergovernmental Panel on Climate*

*Change (IPCC) (IPCC, 2021). SLR projections have considerable uncertainties regarding the timing of core processes, which incurs in different SLR estimations at distinct time periods despite the same global temperature increases (Dewi Le Bars, 2018; IPCC, 2021). Consequently, we explore local estimations of SLR for NYC under a global 2ºC warming at different time periods and considering only processes for which projections can be made with at least medium confidence (IPCC, 2021). The estimations result from multi-model projections with a global mean temperature increase between 1.75ºC and 2.25ºC in 2080-2100 with respect to pre-industrial levels…"*

*Line 308: "We assume temporal independence between the climate scenarios and the SLR scenarios because it allows us to explore more (yet plausible) scenarios. However, previous studies have found that assuming independence between SLR and TCs can underestimate flood hazard (Lockwood et al., 2022)"*

4. Section 2.3.1: Do you control for the timing between the peak surge and peak astronomical tide? As the landfall timing could be different in each of the climate/internal variability scenarios, how do you account for potential differences in the timing of the tide and surge? If the peak surge occurs at low tide, then overall water levels would be lower (but not due to differences in the climate state, just by chance). Also, as you have precipitation maximization scenarios, what about a surge maximization scenario?

*Thank you for the questions. We do not enforce the timing between peak surge and peak astronomical tide because the spectral nudging constraints the timing of the alternative realisations to match the historical event. However, we do double check if the peak surge coincides with the high tides in all runs, which seems to be the case. While there are some minor variations in the peak timing between runs, they all occur during the high tide period (see figure below).*

*While a maximised surge scenario is an interesting idea, optimizing surge levels spatially is more complex and computationally intensive than in the case of precipitation (a single variable). We would need to account for wind speed, mean sea level pressure, and local bathymetry which would affect the local heights along the coast of NYC. For that, we would need to run GTSM repeatedly, which is the most computationally intensive part of our modelling framework. In addition, while the maximised precipitation scenario optimizes precipitation only, the sea level rise scenarios affect strictly the storm surges. This way, while we don't optimize the location of storm surges to make sure they are the highest along the NY coast, we do explore the effects of higher storm surges with the sea level rise scenarios (which makes a balance in how we explore each of the two main drivers with our scenarios: MP for precipitation and SLR for storm surges).*

*We added to the text:*

*Line 150: "We do not explicitly force the timing between peak surge and high tides, as all runs have peak surges occurring within the high tide period (Figure SI XX)."*

[Figure]

Surge levels for alternative realisations of Sandy (coloured lines)

5.  Line 174: Are these water level thresholds based on any impact literature? For example, 2ft (~61 cm) of water is typically considered the point at which most roads become inaccessible (according to US National Weather Service). If the categories can be linked to any rough impact level that would improve the results.

*Thank you for the comment. We link this comment to your comment 12.*

*The categorization of water levels was originally inspired by the study by Koks et al. (2019), where they categorised flood hazards in 4 water levels without making further impact assumptions. We agree that linking the categories to some sort of impact level could improve the relevance of the results. However, we face the complexity of analysing different assets across multiple critical infrastructure systems. These assets are differently affected by floods (say a road against a power station), so we would need to select one type of asset to serve as reference for all categories, as unifying them all under a single impact-inspired metric is challenging and still not very informative. Therefore, we believe that by adopting a generic approach in classifying these thresholds we can still offer valuable information without incurring too much on one or another CI system.*

*Having said that, we agree that 0.05m flooding might not be informative, so we updated the low water level category to 0.15m-0.5m. We chose 0.15m as a value because that's the approximate intermediate height at the "Very low/ no impact" (0.00m – 0.25m) category in Koks et al. (2019). Therefore, our updated categories are: low (0.15m-0.5m), medium (0.5m-1m), and high (>1m).*

*Koks, E.E., Rozenberg, J., Zorn, C. et al. A global multi-hazard risk analysis of road and railway infrastructure assets. Nat Commun 10, 2677 (2019). https://doi.org/10.1038/s41467-019-10442-3*

*The changes in the text are as follows:*

*Line 180: "Different CI assets may exhibit varying responses to distinct flood levels. Unfortunately, comprehensive information regarding the vulnerability of CI assets to specific flood levels is limited (Zio, 2016) and, in particularly, the cost of reconstruction and replacement of CI assets is not available for New York City. Inspired by Koks et al. (2019), we adopt a discrete and qualitative approach by dividing water levels in three categories: low (0.15m-0.5m), medium (0.5m-1m), and high (>1m). This approach allows to quantify the number of exposed assets in each water level category and how it changes under different scenarios, identifying hotspots of impacts, without trying to assign specific monetary value."*

*The new results do not show major differences despite a decrease in the number of exposed assets for the low water level, as seen below:*

*Line 243: "The MP scenario results in the highest number of flooded assets, typically 2.9 times the baseline (Figure 6a). Following this is the SLR101, which shows a 2.2-fold increase, and the SLR71, with a 1.8-fold increase. This is due to the extensive reach of precipitation. As a result, most of the increase in the MP occurs in the low water level category (5.4 times), while SLR71 and SLR101, surge dominated scenarios, increase 1.5 and 1.6 times, respectively."*

[Figure]

6. Section 3.1: Would be helpful to generate an image showing the potential intensity over the North Atlantic during the storm for each scenario. Are there changes in PI due to global warming that are ultimately not translated to the track? Or does the similarity of the tracks stem from similar large-scale TC-favorability conditions?

*Thank you for the suggestion. While potential intensity (PI) is a valuable tool for assessing a tropical cyclone's maximum potential strength, its applicability in our methodology may not be suitable. PI is dependent on SST values, which are different in each climate scenario. However, the differences on SST and PIs are not translated to the storm tracks in our setup. The spectral nudging technique we employ directly aligns the large-scale weather systems with the reanalysis data (NCEP), as explained in line 79. As a result, all simulations share very similar large-scale weather systems, which consequently influence storm tracks, as well as wind speed and mean sea level pressure (MSLP). This is the reason why they are so similar, and why we don't find any signal on these variables. Therefore, even in scenarios with differing SSTs and PI, they should not influence nor explain the variations in storm tracks in our setup.*

*We updated the text to improve the clarity of our experiment:*

*Line 190: "All tracks have landfalls slightly north of the observed landfall, but their mutual differences are minor, and no coherent response of track position to the imposed warming levels is detected, which is to be expected when using spectrally nudged data as the tracks are conditioned by the large-scale weather systems of NCEP (von Storch et al., 2000)."*

*Line 286: "We do not find significant changes for wind speed, MSLP and track position, which could be due to the spectral nudging method where the divergence, vorticity and large-scale weather systems are set to match the reanalysis (von Storch et al., 2000; Weisse and Feser, 2003)."*

7. Lines 183-184: The simulations also underestimate the wind speed at landfall, which would cause underestimation of storm surge. Is the underestimation due to the horizontal resolution of the ECHAM model?

*Thank you for the question. It is true that the wind speed is slightly underestimated during landfall (and more so during the main peak over the Caribbean). According to Hodges et al. (2017), reanalysis products tend to underestimate the peaks in both maximum wind speeds and minimum MSLP (mean sea level pressure). This is likely a consequence of not high enough model resolution and dependence on parameterized processes used in the reanalysis. So, while the model we use underestimates the peaks with respect to the observations, it is still within the range of performance of other modern reanalyses products.*

*Ref: Hodges, Kevin, Alison Cobb, and Pier Luigi Vidale. "How well are tropical cyclones represented in reanalysis datasets?" Journal of Climate 30.14 (2017): 5243-5264.*

*We add to the text:*

*Line 193: "Meteorological features match the observed event well in the region of interest, with some minor underrepresentation of the maximum wind speed (Figure 2b-c)"*

*Line 296: "The simulated storms underrepresent the maximum wind speed and minimum MSLP during the TCs peak over the Caribbean and to a lesser extent during landfall. Similar discrepancies are seen for the other reanalyses tested, indicating the data is within the same range of performance of other reanalyses and models. Peak TC activity is often underrepresented in reanalyses due to limited model resolution and dependence on parametrised processes (Hodges et al., 2017). ECHAM6.1 has an approximate resolution of 0.5 degrees, and studies have shown that*

*higher horizontal resolutions lead to better modelled TCs (Knutson et al., 2020), higher surge heights (Bloemendaal et al., 2019) and better reproduction of precipitation extremes (Prein et al., 2016). However, spectrally nudged model simulations do resolve TC's better than free running simulations (Feser and Barcikowska, 2012; Schubert-Frisius et al., 2017)."*

*In addition, we also added to Figure A3 the range of the spectrally nudged storylines for a better comparison with the reanalyses:*

[Figure]

8. Line 190: I'm surprised there is no change in rainfall at the study area. Just by the Clausius Clapeyron relation one would expect to see an increase in rainfall associated with a 2C increase in mean global temp. Also, previous work by Liu et al. (2018) projected a large increase in rainfall from extra-tropical transitioning TCs under future warming. I understand that the specific spatiotemporal conditions during the storm may not reflect mean projections, but I think the authors need to add some discussion/comparison with previous studies. Also, as I mentioned earlier, figures and discussion about the differences in the large-scale conditions stemming from each climate scenario are needed. That way we can understand how changes (or lack thereof) in the regional conditions affect the features of Sandy's track

*Liu, M., Vecchi, G. A., Smith, J. A., & Murakami, H. (2018). Projection of landfalling-tropical cyclone rainfall in the eastern United States under anthropogenic warming. Journal of Climate, 31(18), 7269– 7286. https://doi.org/10.1175/JCLI-D-17-0747.1*

***Thank you for the comment. We reviewed the precipitation values of Sandy over the study area for the different global warming levels. Indeed, the simulations over the study area do show some relative change in precipitation: the ensemble mean increase from PI to PD is 4%, and the ensemble mean increase from PD to 2C is 9%. This suggests an increase in precipitation also over the study area. However, these changes in mean values could still be the result of internal climate variability: the absolute changes are substantially smaller than during the peak phase of Sandy and there is no clear distinction between the timeseries of each global warming level over the study area (as seen in the figure below at the grey part). Our setup with 3 ensemble members for each global warming level also prevents us from making any robust assessment about this signal and to make confident event attribution to climate change.***

[Figure]

*In addition to the updated results, we believe that some aspects could explain a not very clear increase in the precipitation volume of Sandy over the study area:*

1) *a single event, Sandy, not necessarily following the mean climate trends. Other studies also found diverging conclusions for future versions of Sandy: Lackman (2015) and Gutmann et al. (2018) found a decrease in minimal central pressure, but Yates et al. (2014) did not. Yates et al. (2014) and Gutmann et al. (2018) found no change in wind speed.*

2) *the precipitation rates of Sandy during landfall (our study area) are substantially lower than during the peak over the Caribbean (see Figure 2 d). Gutmann et al. (2018) have shown that most of the precipitation increase occurs during extreme precipitation rates.*

3) *our highest climate scenario is 2C (or 1.6C if you consider the aerosol levels used in that scenario). In fact, other works, such as Yates et al. (2014), have also suggested that the increase in precipitation for Sandy during landfall would occur mostly at higher temperature levels, such as +4C, while at +2C changes are mild (they do not explicitly discuss that response of precipitation to temperature levels, but there are figures showing the response). The other study detecting increase in precipitation for Sandy also explored a warmer scenario than ours (3°–6°C).*

4) *the model resolution being limited to roughly 0.5 degrees. This could lead to underrepresentation of local extremes.*

*Ref:*

*Gutmann, E. D., Rasmussen, R. M., Liu, C., Ikeda, K., Bruyere, C. L., Done, J. M., Garrè, L., Friis-Hansen, P., and Veldore, V.: Changes in Hurricanes from a 13-Yr Convection-Permitting Pseudo–Global Warming Simulation, Journal of Climate, 31, 3643 – 3657, https://doi.org/https://doi.org/10.1175/JCLI-D-17-0391.1, 2018*

*Lackmann, G. M.: Hurricane Sandy before 1900 and after 2100, Bulletin of the American Meteorological Society, 96, 547–560, https://doi.org/10.1175/BAMS-D-14-00123.1, 2015*

*Yates, D., Luna, B. Q., Rasmussen, R., Bratcher, D., Garre, L., Chen, F., Tewari, M., and Friis-Hansen, P.: Stormy Weather: Assessing Climate Change Hazards to Electric Power Infrastructure: A Sandy Case Study, IEEE Power and Energy Magazine, 12, 66–75, https://doi.org/10.1109/MPE.2014.2331901, 2014*

*With all that said, we decided to update the text to mention the increase in mean precipitation across the global warming levels over the study area, the limitations of these changes, and why we still decide to consider them plausible alternative realisations of Sandy without making direct attribution of global warming levels (thus exploring internal climate variability):*

*Line 198: "When averaged across each climate scenario, MSLP and maximum wind speeds show no significant changes (Figure 2b-c). Peak precipitation rates during October 24th to 26th show gains due to climate change (Figure 2d where PI members (blue) lie below the 2C members (brown) for day 24-26), with an ensemble mean 14% increase from PI to PD and a 5% increase from PD to 2C. During landfall over the study area, the ensemble mean increase from PI to PD is 4%, and the ensemble mean increase from PD to 2C is 9%. However, the absolute changes are substantially*

*smaller than during the peak period and there is overlapping between the different climate scenarios (shaded area in Figure 2d). Increases in precipitation generally occur for the most extreme precipitation rates (Gutmann et al., 2018), but by this point, the storm is transitioning into an extratropical (ET) storm, resulting in overall lower precipitation intensity. Thus, we detect some potential climate change signals over the study area, but these signals may also be the result of internal variability. Notably, the considerable differences in the simulations, both across climate scenarios and within them, make them important for further investigation. We therefore focus our subsequent analyses on exploring this variability and its impacts on the study area, without attributing changes to climate change."*

*Line 280: "…Sandy becomes wetter during its peak in the Caribbean in warmer scenarios, yet the precipitation increase over the NYC metropolitan area due to climate change is comparatively smaller and could result from internal variability. Previous studies have found a global increase in precipitation for TCs with climate change (Hill and Lackmann, 2011; Patricola and Wehner, 2018; Knutson et al., 2020), for extratropical cyclones (Liu et al., 2018), and specifically for Sandy (Yates et al., 2014; Gutmann et al., 2018)."*

*Line 290: "Conversely, we observe significant differences between the alternative realizations of Sandy during landfall over the study area. Given our study aim of exploring societal impacts in alternative realisations of Sandy, we decided to focus on the internal variability of the simulations during landfall rather than at the direct effects of climate change on the entire storm lifetime."*

9.   Figure 3: add the tracks to b) and c)

*Thank you for the suggestion. We tried adding the tracks, but we believe it ended up adding more noise to the figure, without adding much value. The objective of these plots is to examine the hazards over the study area exclusively and compare the different ways NYC could be impacted. The track position in these plots did not help with this and ends up adding more information to an already busy plot. We hope the reviewer can understand our perspective.*

10.  Figure 4: add the track to b)

*Thank you for the suggestion. We tried adding the tracks, but we believe it ended up adding more noise to the figure, without adding much value. The objective of these plots is to examine the hazards over the study area exclusively and compare the different ways NYC could be impacted. The track position in these plots did not help with this and ends up adding more information to an already busy plot. We hope the reviewer can understand our perspective.*

11.  Line 225: In this case, maximizing the rainfall (which occurred on the left-hand side of the storm) also minimizes the storm surge (as minimal or negative storm surge values are usually observed on the left side of landfalling TCs). This is because winds are pointed away from the coast on the left side of the storm.

*Thank you for the comment. We agree with it and appreciate the suggestion that improves the results. We have now added to that section:*

*Line 237: "The increased flood volumes due to the MP scenario occur extensively across the study area, but coastal areas show a moderate decrease in inundation levels (Figure 5b). This is due to the MP scenario setup, where inland precipitation over the study area is prioritized. In most realisations of Sandy, the heaviest precipitation occurs on the left-hand side of the storm, which is also characterized by winds blowing away from the coast, resulting in lower surge levels when moved over the study area."*

12. Line 229 "The MP scenario results in the highest number of flooded assets...": But how much of this increase is locations with very low inundation (i.e. on the order of 0.05 m)? I feel that such a low impact threshold of 0.05 m somewhat inflates/exaggerates the impacts of rainfall. Would 0.05 m of water actually cause damage to any building? I assume that most buildings are elevated more than 0.05 m from the bare earth surface

*Thank you for the comment. This question is related to question 5, so the answers are also interconnected. While we do not want to make too many inferences on the impact side because of the limitations explained on question 5, we also agree that it is a valid point that 0.05m might be too low. So, we updated the low water level category to 0.15m-0.5m. We chose 0.15m as a value because that's the approximate intermediate height at the "Very low/ no impact" (0.00m – 0.25m) category in Koks et al. (2019). Therefore, our updated categories are: low (0.15m-0.5m), medium (0.5m-1m), and high (>1m).*

*The changes in text are:*

*Line 180: "Different CI assets may exhibit varying responses to distinct flood levels. Unfortunately, comprehensive information regarding the vulnerability of CI assets to specific flood levels is limited (Zio, 2016) and, in particularly, the cost of reconstruction and replacement of CI assets is not available for New York City. Inspired by Koks et al. (2019), we adopt a discrete and qualitative approach by dividing water levels in three categories: low (0.15m-0.5m), medium (0.5m-1m), and high (>1m). This approach allows to quantify the number of exposed assets in each water level category and how it changes under different scenarios. This approach already allows us to identify potential hotspots of impact, without trying to assign a specific monetary value to this impact."*

*The new results do not show major differences despite a decrease in the number of exposed assets for the low water level, as seen below:*

*Line 243: "The MP scenario results in the highest number of flooded assets, typically 2.9 times the baseline (Figure 6a). Following this is the SLR101, which shows a 2.2-fold increase, and the SLR71, with a 1.8-fold increase. This is due to the extensive reach of precipitation. As a result, most of the increase in the MP occurs in the low water level category (5.4 times), while SLR71 and SLR101, surge dominated scenarios, increase 1.5 and 1.6 times, respectively."*

[Figure]

a) Flooded CI assets and buildings

b) Median increase in flooded CI systems

c) Power: SLR101 increase

d) Road: MP increase